# Diagnostic Accuracy of a Point-of-Care Immunoassay for Feline Immunodeficiency Virus Antibodies, Feline Leukemia Virus Antigen, and *Dirofilaria immitis* Antigen

**DOI:** 10.3390/v15102117

**Published:** 2023-10-19

**Authors:** Seema Singh, Kristen A. Davenport, Elizabeth Schooley, Anthony Ruggiero, Salam Nassar, Jesse Buch, Ramaswamy Chandrashekar

**Affiliations:** IDEXX Laboratories, Inc., Westbrook, ME 04092, USA; kristen-davenport@idexx.com (K.A.D.); elizabeth-schooley@idexx.com (E.S.); anthony-ruggiero@idexx.com (A.R.); salam-nassar@idexx.com (S.N.); jesse-buch@idexx.com (J.B.);

**Keywords:** feline immunodeficiency virus, feline leukemia virus, Dirofilaria immitis, heartworm, serology, diagnosis, screening

## Abstract

Feline immunodeficiency virus (FIV) and feline leukemia virus (FeLV) are retroviral infections of cats worldwide whose clinical manifestations range from mild to severe disease. In both cases, infected cats can live a long life with proper care and should be managed to prevent infection of other cats. *Dirofilaria immitis*, the nematode that causes heartworm disease, can infect cats in any region where dogs are infected. Though cats are more resistant to infection, clinical diseases in the form of heartworm-associated respiratory disease can cause death. Screening for these infectious diseases enables veterinarians to manage their cases and prevent the spread to other cats. We describe the diagnostic accuracy of a point-of-care immunoassay for FIV, FeLV, and heartworm, compared to reference methods commonly available through reference laboratories to the practicing veterinarian. For FIV, we report 100% sensitivity (95% confidence limits (CL): 96.2–100%) and 97.8% specificity (95% CL: 95.4–99.4%). For FeLV, we report 100% sensitivity (95% CL: 97.7–100%) and 99.2% specificity (95% CL: 97.1–99.9%). And for heartworm, we report 90.2% sensitivity (95% CL: 76.9–97.3%) and 100% specificity (95% CL: 98.3–100%). Veterinarians may expect this performance relative to the reference methods they use for confirmatory serological testing.

## 1. Introduction

Feline immunodeficiency virus (FIV) and feline leukemia virus (FeLV) are important infectious diseases of cats worldwide [1]. Members of the retroviridae family, FeLV and FIV insert a DNA copy of their RNA genome into the DNA of the host cell at the time of cell division [2]. Feline heartworm disease, caused by the nematode *Dirofilaria immitis*, is found in tropical and temperate climates that support its mosquito vector [3,4].

FIV is transmitted between cats primarily when the virus-containing saliva is inoculated into a bite wound during fighting [1]. Transmission has also been reported between queens and their kittens [5,6,7]. After exposure, an ineffective humoral immune response allows for life-long infection [1,8]. Typically, cats infected with FIV have a mild clinical course, but it is possible for infected cats to develop an acquired immunodeficiency syndrome. Survival of FIV-infected cats compared to FIV-uninfected cats can vary dramatically, from no difference to a dramatic difference. For example, two groups of cats in a recent study had contrasting outcomes of 37% and 94% survival at 22 months [2,9]. Beczkowski et al. concluded that the housing and management of infected cats have a substantial effect on survival, with cats in single-cat, stable households having much better survival [9]. The reported seroprevalence for FIV antibodies, which depends on method and patient selection, varies worldwide, from about 4% in North America [10], 7.5% in Northern Europe [11], 13.1% in Southern Europe [12], and 13% in Japan [13].

FeLV is transmitted primarily through close contact between cats, including mutual grooming, the oral–nasal route, bite wounds, or vertical transmission between a queen and her kittens [14]. There are multiple outcomes of FeLV infection, including progressive, regressive, focal/atypical, and abortive. The clinical course for an individual patient is determined by virus-host interactions in the early stages of the disease and subsequent changes in immune regulation, such that cats may convert from one outcome to another [15]. Cats with progressive FeLV have a shorter survival than uninfected or regressively infected cats, but they can have a good quality of life with proper care [16]. Seroprevalence (of antigen) in North America was around 2% in 2022 [17], while recent studies in Europe described FeLV prevalence between 0.3% and 21.2% [18,19]. Seroprevalence of FeLV in Asia–Pacific has been reported as high as 6% [11], with one report in Japan as high as 12.2% [20]. Variations in study populations and testing methods can explain these differences.

*Dirofilaria immitis* is transmitted from an infected animal (primarily dogs and wild carnivores) to a susceptible host by a mosquito vector. Although cats are susceptible to *D. immitis*, the species is considered more resistant to infection than dogs. In cats, only a small proportion of L3 larvae develop into adults. Instead, larvae often die in the subcutaneous tissue early after infection, or adults die in the pulmonary arteries 3–4 months post-infection [21]. Most adult heartworm infections in cats have four or fewer adult worms [22]. Death of the parasite at any stage may lead to severe inflammatory reactions within the lungs and death, a syndrome called heartworm-associated respiratory disease (HARD) [23]. *D. immitis* can be found worldwide in tropical and temperate climates where the warm season is long enough for the larvae to develop within the mosquito intermediate host [3]. Due to climate change, more regions are likely to have environmental conditions that support mosquito vectors. Combined with the increased global travel of pets, the likelihood of *D. immitis* being introduced and propagated in previously non-endemic regions is increasing. Our understanding of feline heartworm seroprevalence is limited; the Companion Animal Parasite Council reported 0.57% seroprevalence in the United States and 1.43% seroprevalence in Canada for 2022 [24]. Another 2022 study found a heartworm antigen prevalence of 1.2% in Rio de Janeiro, Brazil [25]. One recent study found a similar prevalence of heartworm in cats to the prevalence in high-risk dogs when using antigen and antibody testing in parallel [26].

The American Association of Feline Practitioners (AAFP) and the International Society for Feline Medicine recommend testing cats for FeLV and FIV when they are first introduced into a household, before vaccination for either virus, after potential exposure, or when displaying clinical signs of disease [1]. Additionally, AAFP recommends all kittens up to a year of age should be tested for FeLV, and negative kittens should be vaccinated. Routine screening for heartworm infection allows clinicians to understand the risk of disease in their area and may help support the routine use of preventatives.

A variety of diagnostic tools exist for each disease, including in-clinic rapid tests, reference laboratory testing, and methods used primarily for research purposes. Each disease presents its own challenges with diagnostic testing and caveats around interpretation. For example, FIV diagnosis is usually serological; antibodies against FIV antigens are considered diagnostic. However, cats generate antibodies against different FIV antigens (for example, Gag vs. Env antigens) at different time points and different levels over the course of the disease. The envelope protein gp40 (transmembrane (TM) protein) is considered the immunodominant protein in FIV infection [27,28,29,30]. Additionally, some anti-Gag antibodies, including those against p15 and p24, are perceived to be less specific for FIV and may be the result of infection with other viruses [31].

Unlike FIV, diagnosis of FeLV is best accomplished by detecting the p27 core protein of the virus itself. PCR is routinely used as well, and in a recent report, 85% of cats have concordant results in PCR and p27 ELISA [32]. However, a combination of physiology and test sensitivity explains the equal populations of cats that are PCR-positive and p27-negative or vice versa [32]. Some cats have very low antigen shedding or antigen shedding localized to lymphoid tissues. These cats have low proviral load and low antigen shedding, increasing the chances of undetectable levels of p27 antigen and this pattern, which has been associated with improved outcomes [33].

Finally, heartworm diagnosis in cats is difficult because cats often have a low worm burden; the antigen detected by most diagnostic tools is shed by adult female worms, but many heartworm infections in cats are arrested before the parasite reaches adulthood (the L5 phase) or may result in a single adult male worm [26]. Detection of antibodies to *D. immitis* is also unreliable, as some asymptomatic cats do not mount an antibody response, and the presence of antibodies may indicate prior infection [26,34]. Detection of heartworm antigen is specific: a cat with circulating *D. immitis* antigen is actively infected. 

Recent updates to the IDEXX SNAP^®^ Feline Triple^®^ point-of-care (POC) immunoassay for the detection of FIV antibody, FeLV antigen, and heartworm antigen allow for an improved customer workflow. Here, we describe the diagnostic accuracy of the updated assay. This level of agreement reflects what a veterinarian can expect when they compare their POC immunoassay results to tests commonly available in reference laboratories.

## 2. Materials and Methods

Sample source. Feline serum samples, originally submitted for clinical testing, were obtained from a commercial diagnostic laboratory (IDEXX Laboratories, Westbrook, ME, USA) in accordance with laboratory terms and conditions and were stored frozen (−20 °C) prior to testing. Presumed positive samples for FIV were banked based on immunofluorescence antibody test (IFAT), PCR, ELISA, or Western blot results. Presumed positive samples for FeLV were banked based on IFAT, PCR, or ELISA results at IDEXX reference laboratories. Presumed positive samples for feline heartworm were banked based on antigen ELISA or antibody testing. Presumed negative samples were submitted for general chemistry and were screened on the on-market SNAP Feline Triple (SNAP^®^ Feline Triple^®^, IDEXX Laboratories, Inc. USDA Product Code 502C.01) according to manufacturer’s instructions before being subjected to reference method testing and POC immunoassay testing (IDEXX Laboratories, Westbrook, ME, USA). The target number of samples was based on sample size calculations designed to minimize the width of the confidence intervals.

Reference method testing. For all reference methods indicated below, a positive and negative control sample appropriate for each assay was used. Serum samples with sufficient volume were randomly selected from the banks described above and tested by the corresponding reference method to classify the samples as positive or negative. For FeLV and heartworm, the reference method and POC immunoassay testing were completed within one week of each other for every sample. For FIV, the Western blot and POC immunoassay were performed within a month for every sample.

FIV testing. A Western blot reagent pack labeled and sold to reference laboratories (FIV Plus Western Blot, IDEXX Laboratories, Westbrook, ME, USA) was used in accordance with the manufacturer’s protocol for the detection of FIV antibodies. Briefly, 10 µL of each serum sample was diluted 1:100 in 1% BSA, 30% FBS, and 0.1% Tween-20 in PBS and incubated with a strip of nitrocellulose that was pre-spotted with FIV antigens for two hours at room temperature. Next, the strip was washed three times for five minutes, rocking at room temperature, with PBS-based wash buffer. A total of 1 mL of 1:6000 HRPO-conjugated anti-cat secondary antibody was added to the strip and incubated for one hour at room temperature. The strip was washed three times for five minutes with the same PBS-based wash buffer. HRP substrate (Bio-Rad, Hercules, CA, USA, #1706431) was prepared according to manufacturer’s instructions and incubated with the strip for 30 min at room temperature. The strip was washed with deionized water to stop development, and all residual liquid was removed from the incubation tray. Bands were observed visually, and results were recorded. The Western blot was classified as positive if a sample had antibodies reacting to *Env* targets, with or without antibodies reacting to *Gag* targets [35]. The Western blot was deemed negative if the sample demonstrated reactivity to zero or only one Gag protein.

FeLV testing. A proprietary microtiter plate ELISA (PetChek^®^ FeLV test, Feline Leukemia Virus Antigen Reagent Pack) developed by IDEXX Laboratories, Westbrook, ME, and used in IDEXX reference laboratories was used for the detection of FeLV antigen in accordance with the manufacturer’s protocol [36]. This protocol has two parts, screening and confirmation by p27 neutralization. Both use inactivated FeLV particles diluted in FBS as a positive control and neat FBS as a negative control. Briefly, 50 µL of each serum sample was mixed with 50 µL mouse monoclonal anti-FeLV antibody conjugated to HRP and added to a microtiter plate coated with another mouse monoclonal anti-FeLV antibody. The plate was incubated at room temperature for 15 min and then washed five times with a PBS-based wash. A total of 100 µL of 3,3′,5,5′-tetramethylbenzidine (TMB) substrate was added to each well and incubated for 15 min. 50 µL stop solution was added to each well, and the absorbance at 650 nm (A_650_) was read with a plate reader spectrophotometer. Samples whose A_650_ ratio relative to the positive control of ≥0.3 were considered positive and were confirmed by virus neutralization. Briefly, two 50 µL aliquots of each serum sample were mixed with 10 µL of either solution A (nonspecific antibody and detergent) or solution B (neutralizing reagent; unlabeled, non-murine, polyclonal anti-FeLV antibody) and incubated for 15 min. 50 µL of those solutions are added consecutively (A, then B) to a new microtiter plate, and the protocol is repeated as described above. Samples are confirmed positive if the B well has absorbance ≥50% of the A well; in other words, p27 neutralization must reduce the signal by greater than or equal to 50% for the sample to be considered positive. If the A well, which contains nonspecific antibodies, has a signal higher than the positive control well and its corresponding B well does not have ≥50% signal reduction, then the sample must be diluted. In this case, samples are diluted 1:10 in solution A (the nonspecific antibody), and the p27 neutralization protocol is repeated with 50 µL of diluted sample.

*Dirofilaria immitis testing.* A proprietary microtiter plate ELISA (Heartworm Antigen by ELISA) developed by IDEXX Laboratories, Westbrook, ME, and used in IDEXX reference laboratories was used for the detection of *D. immitis* antigen in accordance with the manufacturer’s protocol. Briefly, 100 µL of the sample was added to a microtiter plate coated with anti-heartworm antibody and incubated for thirty minutes at room temperature. Wells were washed five times with a PBS-based wash buffer. A total of 100 µL of anti-heartworm antibody conjugated to HRP was added to each well and incubated for thirty minutes at room temperature. The same wash procedure was repeated, and 50 µL TMB substrate was added to each well and incubated for ten minutes at room temperature. 50 µL of stop solution was added to each well, and the A_650_ was recorded. Samples were considered positive if absorbance was higher than a cutoff based on the negative control. 

*POC immunoassay testing.* All serum samples were blinded and randomized before being tested using the POC immunoassay according to the manufacturer’s instructions (SNAP^®^ Feline Triple^®^, IDEXX Laboratories, Inc. USDA Product Code 502C.02). These test devices use bidirectional flow of the test sample, wash, and substrate along with enzymatic amplification to produce visible blue spots when test results are positive. Briefly, 150 µL of serum was added to a tube with four drops (200 µL) of conjugate reagent, which contains HRP-conjugated anti-FeLV antibody, FIV antigen, and anti-heartworm antibody. This solution was mixed and immediately added to the sample well. Sample was allowed to flow down the membrane until flow reached the activation circle, at which point the device was activated by snapping. The assay was allowed to develop for ten minutes at room temperature, and the results were read visually. 

In this case, there are three independent assays: FeLV antigen, FIV antibody, and heartworm antigen, which are multiplexed on one device, with different locations for each assay and corresponding visible spot. Three individuals blinded to the sample source read each test device. A consensus for visual interpretation required that at least 2/3 of operators independently read a positive or a negative color reaction for the result to be recorded as positive or negative. 

The FeLV assay detects p27 antigen [36], the FIV assay detects antibodies against the immunodominant transmembrane protein gp40 [28,29,30], and the heartworm assay detects circulating antigen from D. immitis [37].

*Statistical analysis.* Sensitivity and specificity were calculated for each POC immunoassay relative to the corresponding reference methods by dividing the number of samples that were positive on the POC immunoassay by the number that were positive on the reference method (sensitivity) and the number of samples that were negative on the POC immunoassay by the number of samples that were negative on the reference method (specificity). The determination of Clopper Pearson 95% confidence intervals was performed using R (version 4.1.0) [38].

## 3. Results

### 3.1. FIV

A total of 279 serum samples were tested. 95 samples were classified as positive for FIV antibodies based on the FIV Plus Western blot. All 95 were classified as positive on SNAP Feline Triple. A total of 184 samples were classified as negative based on the FIV Plus Western blot. Of those, 180 samples were classified as negative on SNAP Feline Triple (Table 1). Therefore, the FIV sensitivity is 100% (95% confidence limits 96.2–100%), and the FIV specificity is 97.8% (95% confidence limits 95.4–99.4%).

### 3.2. FeLV

A total of 407 serum samples were tested. A total of 158 samples were classified as positive for FeLV p27 antigen on the PetChek FeLV ELISA. All 158 samples were classified as positive on SNAP Feline Triple. A total of 249 samples were classified as negative on the PetChek FeLV ELISA. Of those, 247 samples were classified as negative on SNAP Feline Triple (Table 2). Therefore, the FeLV sensitivity is 100% (95% confidence limits 97.7–100%), and the FeLV specificity is 99.2% (95% confidence limits 97.1–99.9%). 

### 3.3. Heartworm

A total of 256 serum samples were tested. A total of 41 samples were classified as positive for D. immitis antigen on the PetChek Heartworm ELISA. Of those, 37 were classified as positive on SNAP Feline Triple. A total of 215 samples were classified as negative on the PetChek Heartworm ELISA. Of those, 215 samples were classified as negative on SNAP Feline Triple (Table 3). Therefore, the heartworm sensitivity is 90.2% (95% confidence limits 76.9–97.3%) and the heartworm specificity is 100% (95% confidence limits 98.3–100%). 

## 4. Discussion

The updated SNAP Feline Triple has high specificity for all three infections (FIV, FeLV, and heartworm) and excellent sensitivity for FIV and FeLV, with acceptable sensitivity for heartworm infection. The test’s performance is comparable to previous versions of the SNAP Feline Triple and SNAP Feline Combo products [39,40]. Therefore, SNAP Feline Triple serves as a suitable in-clinic screening test for cats living in or with a travel history to heartworm endemic areas. SNAP Feline Triple is especially valuable in the context of the greater FIV/FeLV/heartworm testing paradigm; the rapid test, with its excellent specificity, can be used as a primary screening tool. The veterinarian has multiple options for confirmatory testing in the case of positive results, including ELISA and PCR testing available at commercial laboratories for FIV and heartworm and quantitative real-time PCR and ELISA for FeLV. 

As with any diagnostic performance evaluation, the choice of reference method affects the apparent performance. For this study, we chose reference methods available through reference laboratories to practicing veterinarians rather than more traditional reference methods such as virus isolation or necropsy so that the performance in the study would reflect the performance experienced in clinical practice. Western blot is generally accepted as the reference method for FIV diagnosis, though it has caveats. Since Western blot for FIV antibody detection was originally proposed, experts have used different banding patterns to classify a sample as positive. Hosie recommended that a sample be classified as positive if antibodies against Env *or* at least three Gag proteins are detected [35]. Others have used the criteria of *any two bands* on a Western blot to call a sample positive [41,42]. Thus, our inclusion criteria are similar to Hosie and Jarrett’s requirement for *Env* reactivity. We used ELISA as the gold standard for FeLV diagnosis; p27 antigenemia is correlated with proviral loads, but either antigen concentration or proviral loads can fall below the limit of detection, causing infrequent discrepant results (~14%) [32]. Necropsy is still considered the gold standard for heartworm diagnosis. Of course, necropsy is not a practical means of confirming a diagnostic result. Our choice to use an ELISA as the reference method means that we have evaluated the sensitivity and specificity of SNAP^®^ Feline Triple to detect *heartworm antigen*, which will include cats with heartworm infections that include the adult female worms that shed antigens. The excellent specificity means that cats with a positive test result are very likely to be truly infected. The SNAP test missed four cats that were positive on ELISA for heartworm antigen; these discrepant results are most likely due to the lower limit of detection by the ELISA assay.

This study was designed to demonstrate the analytical performance of the POC test. However, the interpretation of diagnostic results by clinicians is informed by clinical presentation and history, physical exam findings, other diagnostic test results, and the clinician’s judgment, not just the test results. The sensitivity and specificity of all diagnostic results are less important than the positive and negative predictive value, which is influenced by the prevalence in the population which is based on the clinician’s choice of which animals to test. Our study is limited by the simplified measurement of sensitivity and specificity without considering the clinical picture of the cats being tested. Additionally, we limit our diagnostic testing to serology to understand the performance of a serological POC test. However, for FIV and FeLV, the clinician has the option to add PCR to better understand the infection status of the cat.

Based on AAFP guidelines, cats should be screened for FeLV and FIV when they are new to the home (including kittens for FeLV), after exposure to either virus, prior to vaccination or anytime they present with clinical illness. Additionally, blood donors should be screened for these viruses (both antigen and PCR testing) to ensure the donor will not transmit the viruses to the blood recipient [43]. Although heartworm testing is not required prior to starting preventatives in cats, routine screening reveals to the veterinarian the prevalence of heartworm in their region, may support the routine use of preventatives in areas previously considered non-endemic, and illustrates the climate-related spread of the mosquito vectors that enable heartworm endemicity. 

In conclusion, the updated SNAP Feline Triple provides excellent sensitivity and specificity for FeLV/FIV infection and acceptable sensitivity with excellent specificity for heartworm antigen. 

## Figures and Tables

**Table 1 viruses-15-02117-t001:** **FIV diagnostic accuracy**. Performance of SNAP Feline Triple compared to the Western blot reference method with positive and negative samples. Sensitivity and specificity point estimates and 95% confidence limits are calculated.

		FIV Plus Western Blot		
		Positive	Negative	Total
SNAP Feline Triple FIV	Positive	95	4	99
	Negative	0	180	180
	Total	95	184	279
**Sensitivity**	*95% CL*	**100%**	*96.2–100%*	
**Specificity**	*95% CL*	**97.8%**	*95.4–99.4%*	

**Table 2 viruses-15-02117-t002:** **FeLV diagnostic accuracy**. Performance of SNAP Feline Triple compared to the ELISA reference method with positive and negative samples. Sensitivity and specificity point estimates and 95% confidence limits are calculated.

		PetChek FeLV ELISA		
		Positive	Negative	Total
SNAP Feline Triple FeLV	Positive	158	2	160
	Negative	0	247	247
	Total	158	249	407
**Sensitivity**	*95% CL*	**100%**	*97.7–100%*	
**Specificity**	*95% CL*	**99.2%**	*97.1–99.9%*	

**Table 3 viruses-15-02117-t003:** **Heartworm diagnostic accuracy**. Performance of SNAP Feline Triple compared to the ELISA reference method with positive and negative samples. Sensitivity and specificity point estimates and 95% confidence limits are calculated.

		PetChek Heartworm ELISA		
		Positive	Negative	Total
SNAP Feline Triple HW	Positive	37	0	37
	Negative	4	215	219
	Total	41	215	256
**Sensitivity**	*95% CL*	**90.2%**	*76.9–97.3%*	
**Specificity**	*95% CL*	**100%**	*98.3–100%*	

## Data Availability

Not applicable.

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
