# Peer review of "Diagnostic Accuracy of a Point-of-Care Immunoassay for Feline Immunodeficiency Virus Antibodies, Feline Leukemia Virus Antigen, and Dirofilaria immitis Antigen"

_viruses, 2023, doi:10.3390/v15102117_

Round 1
Reviewer 1 Report
I think there was some inappropriate self-citation from one of the authors – why cite an abstract about FIV/FeLV when there are at least 10 or 20 published papers which would be much better references? (Buch, J., Beall, M., O'Connor, T., & Chandrashekar, R. (2017). Worldwide Clinic-Based Serologic Survey of FIV Antibody 316 and FeLV Antigen in Cats. ACVIM, (p. ID14))
Minor comments below if the authors consider resubmitting at another journal.
ABSTRACT:
Change ‘we find’ to ‘we report’
L16 – state these results relate to testing serum
L16-17 – make it clear the reference test used was IDEXX reference tests – i.e. ‘to reference methods available through IDEXX reference laboratories’
INTRODUCTION:
‘Average survival of infected cats is not different from survival in uninfected cats, although survival times vary.’ This statement feels like a disappointing oversimplification of a very difficult area, with one reference provided that I cannot trace because of the non-numbered reference list! Survival time CAN vary – e.g. BÄ™czkowski PM, Litster A, Lin TL, Mellor DJ, Willett BJ, Hosie MJ. Contrasting clinical outcomes in two cohorts of cats naturally infected with feline immunodeficiency virus (FIV). Vet Microbiol. 2015;176(1-2):50-60. Please try and avoid sweeping oversimplified statements.
The prevalence rates provided for FIV (and website used as reference for these rates) are inaccurate and bare. E.g. you confidently report 13% for Japan, yet there are other references that put it at 29% (Ishida T, Washizu T, Toriyabe K et al. Feline immunodeficiency virus infection in cats of Japan. J Am Vet Med Assoc 1989;194(2):221–225) and 23% (Furuya T, Kawaguchi Y, Miyazawa T et al. Existence of feline immunodeficiency virus infection in Japanese cat population since 1968. Jpn J Vet Res 1990;52(4):891–893). These prevalence numbers also heavily rely on study design (as you highlight for FeLV in L54-55). Since this is a UC-centric study – presumably all the testing was done at IDEXX Laboratories in the US (which also should be noted in the manuscript) – I would suggest the authors to simply focus on US prevalence rates for all diseases. Why only highlight Japan and the US for FIV? Alternatively, you can add a few more numbers and study references for global FIV prevalence, as you have done for FeLV prevalence.
L46, 49 – they are not different ‘stages’ of FeLV infection – they are different outcomes following exposure. Please change
L93-95 – Please list the anti-gag antibodies here – p10, p15, p24?
METHODS
Can you please clearly state the targets for each assay in the PoC kit, and generalities around how the testing has improved from previous versions?
L119 -122 – Need to state how long these samples may have been stored for. Long term storage at -20 degrees can affect sample quality
L122 – Where was testing performed?
L123-124 – ‘based on immunofluorescence antibody test (IFAT), PCR, ELISA, or Western blot results’ – performed at IDEXX?
L149-173 – virus neutralization technically involves live virus assays (i.e. cell culture). Do you mean antigen p27 neutralization here?
RESULTS
L210 – FIV = 279 samples. L217 – FeLV = 407. L224 – HW = 256 samples. Can you justify these numbers? Was this just what was collected over a certain time period?
Author Response
Please see the attached response to reviewer 1.

Reviewer 2 Report
this study describes the diagnostic accuracy of a point-of-care immunoassay for FIV, FeLV and heartworm, compared to reference methods available.
Feline serum samples were obtained from a commercial diagnostic laboratory and presumed positive samples were banked based on IFAT, PCR, ELISA, or Western blot results.
Authors should describe the diagnostic performances of these test , as well as of the reference methods (Se and SP).
How many presumptive positive and negative samples, for each test, were used ? Presumptive negative samples were tested negative by FAT, PCR, ELISA, or Western blot ?
Author Response
Please see the attached response to reviewer 2.

Reviewer 3 Report
The paper can be published after some important adding and clarification (see attached PDF).
The Authors must clarify the rationale of this study (and of this publications) since their results are comparable of those of the same existing tests. Has anything changed from the past?

An extensive English revision is needed since some parts are not easy to read and understand.
Author Response
Please see the attached response to reviewer 3.

Round 2
Reviewer 1 Report
Well done to the authors on a professional and high quality reply which I enjoyed reading. I have only a few minor comments below.
L38-40 - Please consider rephrasing this sentence from:
Average survival of infected cats is not different from survival in uninfected cats, 38 although survival times of infected cats can vary dramatically (two groups of cats in a 39 recent study had 37% or 94% survival at 22 months [2,9].
To
Survival of FIV-infected cats compared to FIV-uninfected cats can vary dramatically, from no difference to a dramatic difference. For example two groups of FIV-infected cats in a recent study had very contrasting outcomes of 37% and 94% survival at 22 months [2,9].
L211-212 'the FIV assay detects antibodies specific to FIV' - please specify what the target antigens used for FIV antibody detection are here. p15 and p24?
